# Childhood Apraxia of Speech: A Descriptive and Prescriptive Model of Assessment and Diagnosis

**DOI:** 10.3390/brainsci14060540

**Published:** 2024-05-24

**Authors:** Ahmed Alduais, Hind Alfadda

**Affiliations:** 1Department of Human Sciences (Psychology), University of Verona, 37129 Verona, Italy; 2Department of Curriculum and Instruction, King Saud University, Riyadh 11362, Saudi Arabia

**Keywords:** apraxia of speech, assessment model, childhood apraxia, cultural diversity, neurobiological markers, speech motor control

## Abstract

Childhood apraxia of speech (CAS) represents a significant diagnostic and therapeutic challenge within the field of clinical neuropsychology, characterized by its nuanced presentation and multifactorial nature. The aim of this study was to distil and synthesize the broad spectrum of research into a coherent model for the assessment and diagnosis of CAS. Through a mixed-method design, the quantitative phase analyzed 290 studies, unveiling 10 clusters: developmental apraxia, tabby talk, intellectual disabilities, underlying speech processes, breakpoint localization, speech characteristics, functional characteristics, clinical practice, and treatment outcome. The qualitative phase conducted a thematic analysis on the most cited and recent literature, identifying 10 categories: neurobiological markers, speech motor control, perceptual speech features, auditory processing, prosody and stress patterns, parent- and self-report measures, intervention response, motor learning and generalization, comorbidity analysis, and cultural and linguistic considerations. Integrating these findings, a descriptive and prescriptive model was developed, encapsulating the complexities of CAS and providing a structured approach for clinicians. This model advances the understanding of CAS and supports the development of targeted interventions. This study concludes with a call for evidence-based personalized treatment plans that account for the diverse neurobiological and cultural backgrounds of children with CAS. Its implications for practice include the integration of cutting-edge assessment tools that embrace the heterogeneity of CAS presentations, ensuring that interventions are as unique as the children they aim to support.

## 1. Introduction

### 1.1. Introduction to Childhood Apraxia of Speech

Childhood apraxia of speech (CAS) is a complex and multifaceted speech sound disorder characterized by difficulties in the acquisition, production, and perception of speech. This condition is not attributable to neuromuscular deficits but involves impaired planning and consistency of speech movements [1]. The prevalence of CAS decreases with age, but early diagnosis is crucial, as persistent cases are more likely to co-occur with neurodevelopmental disorders, language impairments, and reading difficulties [2]. CAS is more commonly diagnosed in boys, with gender differences in prevalence diminishing by school age [3].

The diagnostic process for CAS is nuanced and multifaceted, often involving the expert judgment of a speech–language pathologist (SLP) alongside various assessment tools. Currently, there are no standardized screening instruments specific to CAS, making the diagnosis sometimes reliant on observations made throughout treatment rather than at an initial screening [4]. The presentation of CAS can include difficulties with the motor skills that utilize facial musculature, such as chewing and blowing one’s nose, indicating a broader motor coordination impairment which may require a multidisciplinary approach for assessment and intervention [3]. The following Section presents an overview of definitions of CAS, its neurological foundations, and diagnostic and assessment measures.

To fortify the theoretical framework concerning the diagnosis and assessment of childhood CAS, it is pivotal to incorporate findings from systematic reviews and meta-analyses into our discourse. Chilosi et al. conducted a systematic review to discern discriminative features for distinguishing CAS from other speech sound disorders (SSDs), revealing a dire need for high-quality diagnostic studies and a diverse range of discriminative measures which require further validation and replication [5]. Similarly, Oliveira et al. systematically reviewed methods assessing CAS, pointing to the necessity of validating assessment protocols which effectively integrate motor, articulatory, and segmental evaluations [6]. Murray et al. added a multidimensional perspective by illustrating how CAS interacts with comorbid neurodevelopmental disorders, emphasizing the complexity of CAS profiles and the need for personalized and multidimensional diagnostic approaches [7]. These studies collectively underscore the urgency of developing robust, validated diagnostic tools and assessment protocols grounded in comprehensive systematic research to enhance accuracy and reliability in diagnosing CAS. Integrating these insights will ensure that our discussion is well documented and meticulously connected, thereby enriching our understanding and approaches to managing this complex speech disorder.

### 1.2. Defining Childhood Apraxia of Speech

CAS stands as a distinct neurological speech sound disorder characterized by the intricate breakdown of precision and consistency in the movements underlying speech, without neuromuscular deficits [8]. The disorder’s complexity is underscored by its association with a spectrum of impairments spanning from phonological awareness, language, and literacy [9] to more pervasive cognitive functions, including nonverbal sequential functioning [10]. Despite its multifaceted presentation, CAS is commonly diagnosed based on core features: inconsistent error production on consonants and vowels, disrupted coarticulatory transitions, and inappropriate prosody [1,11].

The challenges inherent to CAS diagnosis and treatment are compounded by the disorder’s rarity and the ongoing debate over its underlying causes. Intervention studies, such as those examining the Nuffield Dyspraxia Programme-3 and the Rapid Syllable Transition Treatment, offer evidence of the efficacy in improving speech production, yet also highlight the requirement for a multifaceted approach to treatment which addresses the broad scope of impairments associated with CAS [12]. Moreover, the empirical evidence underscores the necessity for early intervention strategies, such as the introduction of sign language, which have been shown to facilitate rapid improvement in language development [13].

Current research advocates for a diagnostic process that not only accounts for overt speech impairments but also considers the potential role of genetic factors and neurobiological correlates [5,14]. However, the heterogeneity of CAS, as evidenced by its variable impact on speech perception abilities [15], necessitates diagnostic and treatment approaches that are individualized and responsive to the broad range of speech and language difficulties presented by children with this disorder.

### 1.3. Neurological Foundations of Childhood Apraxia of Speech

CAS is recognized as a neurogenic speech sound disorder with a complex neurological underpinning. The disorder has been described as a deficit in the programming of sequential articulatory movements, absence of neuromuscular deficits, indicating a breakdown in the motor planning and coordination required for speech production [16,17]. The neurological basis of CAS is further substantiated by studies that have observed focal degeneration in superior premotor and supplementary motor areas, areas critical to speech production, in individuals with primary progressive apraxia of speech [18,19]. This degeneration is consistent with the speech motor planning and programming deficits observed in CAS, cementing the disorder’s classification as a neurological syndrome.

The assertion that CAS is a neurological disorder is bolstered by findings that link it to specific genetic pathways and subtle brain anomalies. For instance, the discovery of FOXP2 variants and the association of CAS with conditions such as 16p11.2 microdeletion syndrome contribute to our understanding of CAS’s neurobiological etiology [14,20,21]. Moreover, children with CAS have been found to exhibit comorbidities in nonverbal sequential functioning, further implicating neurodevelopmental factors in the disorder’s manifestation [10]. While MRI scans may not reveal overt neural anomalies in individual cases, quantitative MRI methods have unveiled subtle group-level brain anomalies, and multidimensional investigations have confirmed the need for a comprehensive approach to diagnosis and management that considers CAS’s neurobiological roots [11,22]. Therefore, current evidence strongly indicates that CAS is a neurological disorder with a multifaceted presentation, necessitating a multidimensional diagnostic and therapeutic approach.

The Directions into Velocities of Articulators (DIVA) model and the Gradient Order DIVA (GODIVA) model are innovative neurobiologically grounded computational frameworks used to understand speech motor control and sequencing [23]. These models are particularly crucial for studying and treating motor speech disorders such as apraxia of speech (AOS). By outlining how phonological plans are transformed into motor commands and sequenced over time, the DIVA and GODIVA models provide a comprehensive explanation of both the production mechanisms and the potential neural disruptions leading to AOS, facilitating targeted research and therapeutic approaches [23].

The DIVA/GODIVA neurocomputational framework provides a comprehensive model for understanding the apraxia of speech. DIVA_EEG, an extension of DIVA, incorporates electroencephalography to enhance the model’s physiological validation [24]. The GODIVA model, an extension of DIVA, focuses on learning individual sounds and chunking frequently produced phoneme sequences, crucial to speech motor control development in infants [25]. LaDIVA, another extension of DIVA, integrates physiologically based laryngeal motor control, enabling simulations of various laryngeal motor control modes and generating prosodic contours in speech [26]. These models collectively offer a neurobiologically grounded computational framework that can elucidate the complexities of the apraxia of speech, guiding personalized interventions for individuals with speech disorders.

### 1.4. Diagnostic Tools for Childhood Apraxia of Speech

In this study, the term “diagnosis” refers to the clinical or medical determination made to officially identify an individual as having apraxia. CAS is diagnosed through a combination of expert judgment of perceptual features and objective measures. Diagnostic criteria, as outlined by the American Speech–Language–Hearing Association, include inconsistent error production on consonants and vowels, lengthened coarticulatory transitions between sounds, and inappropriate prosody [11]. Tools such as the Pause Marker Index (PMI) have been validated as markers to scale the severity of CAS, demonstrating significant correlations with other signs of CAS precision and stability [27]. Moreover, discriminant function analysis, including measures from polysyllabic picture-naming tasks and oral–motor examinations, has been used to predict expert diagnoses with a high accuracy [28,29].

In the search for more refined diagnostic tools, computational neural modeling has been utilized to hypothesize about underlying deficits, leading to predictions which can be empirically tested [30]. Furthermore, the Apraxia of Speech Rating Scale (ASRS) provides a quantitative measure for the presence, frequency, and severity of AOS characteristics, with strong indices of reliability and validity [31]. Additionally, automated tools like “Tabby Talks” are emerging, which can remotely detect common errors associated with CAS, potentially reducing therapist workload and family costs [32]. Despite these advancements, a critical lack of well controlled treatment studies persists, which hinders the establishment of definitive conclusions regarding the most effective interventions for treating CAS [33]. Hence, while various tools and approaches have been developed to aid in the diagnosis and assessment of CAS, continued research is necessary to consolidate these methods into standardized clinical practice.

### 1.5. Tools for Assessing Childhood Apraxia of Speech

In this study, the term “assess” is used to describe the thorough evaluation of apraxia, which may encompass reports from parents and teachers, as well as informal assessments which are not formally accredited for diagnosing apraxia. Assessing CAS involves a variety of tools that aim to capture the multifaceted nature of the disorder. The gold standard remains the expert judgment of perceptual features, including inconsistent error productions, lengthened coarticulatory transitions, and inappropriate prosody [29]. The Apraxia of Speech Rating Scale (ASRS) offers a standardized method to document the frequency and severity of these speech characteristics, exhibiting high intra- and inter-judge reliability as well as strong correlations with clinical judgments [31]. Additionally, the Dynamic Evaluation of Motor Speech Skill (DEMSS) provides a structured assessment to aid in differential diagnosis, showing promise in distinguishing CAS from other severe speech impairments [34].

Recent advances have seen the development of automated tools such as “Tabby Talks”, which utilizes a speech processing pipeline to detect common errors associated with CAS, potentially streamlining the diagnostic process [32]. Moreover, neurocomputational modeling approaches have been employed to understand the speech motor deficits underlying CAS, which could lead to the generation of testable hypotheses and inform treatment [30]. Despite these advancements, a systematic review highlighted the scarcity of well controlled treatment studies, indicating a need for more robust evidence to support intervention efficacy [33]. Taken together, these tools form a comprehensive arsenal for assessing CAS, combining traditional clinical expertise with innovative technology to enhance accuracy and efficiency in diagnosis.

### 1.6. Rationale

Despite significant advancements in the field of CAS, there remains a notable gap in the development and validation of standardized diagnostic tools that are specifically tailored for early detection. Current methodologies heavily depend on subjective assessments by speech–language pathologists and lack a universally accepted, objective screening instrument for the initial evaluations. This gap is significant, considering the critical importance of early diagnosis for effective intervention and improved outcomes in CAS, as early intervention is crucial for mitigating the associated neurodevelopmental challenges and enhancing language acquisition.

The rationale for our study is to address this deficiency by synthesizing existing diagnostic measures and proposing a refined, standardized toolset that can be reliably used across different clinical settings. By integrating quantitative cluster analysis with qualitative thematic analysis, our research aims to distil and enhance the current diagnostic criteria and assessment procedures. This study, therefore, stands at the confluence of empirical evidence and clinical innovation, charting a course for the progressive articulation of CAS diagnostic and assessment strategies that are both scientifically grounded and pragmatically applicable.

### 1.7. Purpose of the Present Study

The aim of this mixed-method design study is to explore and synthesize the existing diagnostic measures and assessment procedures for CAS. By employing a robust quantitative cluster analysis coupled with a nuanced qualitative thematic analysis, the study seeks to distil the wealth of research delineated within the top-cited and most recent studies in the field. The dual-faceted analytical approach endeavors to uncover the patterns, key themes, and seminal insights that underpin the intricate diagnostic landscape of CAS. This systematic content analysis is not confined to mere *descriptive* ends; it is strategically geared towards *prescribing* a conceptual framework that will serve to refine and enhance the diagnostic and assessment toolkit available to clinicians and researchers alike.

## 2. Methods

### 2.1. Sampling

This mixed-methods study commenced with a quantitative phase involving a systematic search within the Web of Science Core Collection and Scopus databases, yielding a total of 268 and 311 documents, respectively. After the elimination of duplicates using Mendeley, 290 articles remained, forming the basis for the cluster analysis. These articles consisted of full papers published in English that focused on the diagnosis and assessment of CAS. For the qualitative phase, the 25 top-cited documents on CAS—ranging from 164 to 53 citations—and the 10 most recent documents were selected for the thematic analysis. The decision to select this specific number of documents was twofold and strategically considered. Firstly, the constraints of this paper’s length limited the amount of data that could be thoroughly analyzed and synthesized. Secondly, we believed that reviewing the 10 most recent and 25 top-cited documents would provide a representative sample that effectively indicated the prevailing trends and directions within the field. This approach ensured that our analysis remained both manageable and meaningful, capturing the core developments shaping this area of study. Both phases utilized terms such as “childhood apraxia of speech”, “developmental apraxia of speech”, “articulatory apraxia”, “developmental verbal dyspraxia”, and “verbal dyspraxia”, ensuring a comprehensive and representative dataset of the literature, spanning from historical to contemporary and most influential in the field. Th full used search strings are below:
“childhood apraxia of speech” (Title) or “developmental apraxia of speech” (Title) or “articulatory apraxia” (Title) or “developmental verbal dyspraxia” (Title) or “verbal dyspraxia” (Title) or “childhood dyspraxia” (Title)


### 2.2. Design

This study employed a mixed-method design, integrating both cluster and thematic analyses. The quantitative component involved a cluster analysis to discern existing data clusters on CAS diagnosis and assessment. The qualitative segment followed, with an in-depth thematic analysis of the identified top-cited and most recent documents. The culmination of these analyses facilitated the development of a model that not only described but also prescribed the diagnostic and assessment measures for CAS.

### 2.3. Measures

For the quantitative analysis, CiteSpace (Version 6.3.R1) [35] and VOSviewer (Version 1.6.19) [36] were utilized for clustering and data visualization. These tools facilitated the organization and interpretation of the extensive literature on CAS diagnosis and assessment. The qualitative thematic analysis drew upon the insights gained from the quantitative data, focusing on the 25 top-cited and 10 most recent studies. The themes were generated based on the patterns and information distilled from these documents, and the resulting model was designed by integrating key findings from both phases of analysis.

### 2.4. Reliability, Validity, and Trustworthiness

Ensuring the methodological rigor of both the quantitative and qualitative components of this study was paramount. For the quantitative phase, *internal validity* was meticulously verified through a screening process that reviewed both title and abstract relevancy, focusing strictly on the measurability of the targeted topic—the diagnosis and assessment of CAS—and excluding any studies which did not align. The robustness of the search strategy, executed on 1 March 2024, and the careful removal of duplicates provided a representative and comprehensive dataset for analysis. *External validity* was bolstered by the fact that the studies were not confined to a specific presentation of CAS, thus enhancing the generalizability of the findings to the broader spectrum of the disorder, albeit with caution against conflating CAS with other speech disorders like acquired apraxia or dysarthria. The *reliability* of the quantitative data was strengthened through a meticulous documentation of the data collection process, employing standardized and replicable search terms and methodologies. The detailed reporting of cluster analysis procedures, using established software such as CiteSpace and VOSviewer, further reinforced the reliability of the findings. *Objectivity* was maintained throughout the quantitative analysis, ensuring that the data-driven results were a product of empirical evidence rather than researcher bias or influence.

For the qualitative data, *credibility* was achieved by engaging in peer debriefing sessions to foster confidence in the “truth” of the findings. *Transferability* was attained through the provision of thick descriptions of the procedures applied during data collection and data analysis, offering sufficient context for the results to be applicable in other settings. *Dependability* was ensured through external auditing, involving an independent researcher’s examination of both the research process and outcomes, validating the accuracy and supporting the conclusions drawn from the data. *Confirmability*, a testament to this study’s neutrality, was achieved by employing a triangulation method that involved cross-verification among multiple data sources, including the 25 top-cited studies from the Web of Science and the 10 most recent studies from Scopus, to eliminate researcher bias and ensure that the findings were shaped by the data and not by the researchers’ motivations or interests.

### 2.5. Procedure

The search for the relevant literature was conducted meticulously, with articles retrieved from the Web of Science and Scopus databases on 1 March 2024, using specified search terms related to CAS. The retrieved data underwent cluster analysis using CiteSpace and VOSviewer, allowing for the identification and visualization of prevalent research themes and gaps. This preparation laid the groundwork for a more targeted thematic analysis.

The 290 studies for cluster analysis were selected according to their close relevance to the diagnosis and assessment of CAS. Further, the 25 top-cited studies were carefully selected based on their citation count, ensuring the inclusion of the most impactful research in the field. Additionally, the 10 most recent studies were incorporated to provide a current perspective on CAS assessment and diagnosis. These studies, along with 40 additional papers pertinent to CAS (included in Section 1 and Section 4), were rigorously reviewed to ensure their relevance before initiating the thematic analysis. This step was crucial in affirming the studies’ applicability to the research questions at hand. The thematic analysis proceeded with the extraction of data, which were then methodically compared and contrasted by the research team to ascertain their pertinence to this study’s aims. This comprehensive review process, augmented by a stringent search strategy, culminated in the extraction of data presented in Section 3. The final step was the synthesis of the main findings from both the cluster and thematic analyses, leading to the development of a conceptual model for the diagnosis and assessment of CAS.

## 3. Results

The findings of this investigation are delineated in two distinct sections: cluster analysis and thematic analysis. The former section provides an account of the quantitative data through their rigorous analysis and subsequent interpretation, whereas the latter section elucidates the qualitative data, providing a detailed analysis and interpretive commentary on the themes.

### 3.1. Cluster Analysis

Figure 1 is a visualization of the co-occurrence of keywords related to measures of assessment and diagnosis of CAS, generated using the VOSviewer software. Six clusters were identified in the data, each represented by a different color: red, green, yellow, blue, turquoise, and purple. The red cluster focuses on the assessment of speech sound production in CAS, with keywords such as “speech sound disorders” and “prosody.” The blue cluster focuses on the differential diagnosis of CAS. The yellow cluster focuses on the acoustic aspects of CAS, with keywords such as “phonology” and “speech. The turquoise cluster appears to be related to therapy and intervention. The green cluster focuses on the phonological awareness of CAS. The purple cluster is small and includes the keyword “treatment”, suggesting a potential link between the assessment, diagnosis, and treatment of CAS.

Figure 2 shows the top ranked item by bursts in the developmental apraxia of speech in Cluster #0, with bursts of 6.19. The second one is verbal dyspraxia in Cluster #0, with bursts of 5.30. The third is phonological disorders in Cluster #0, with bursts of 4.16. The fourth is speech in Cluster #0, with bursts of 3.85. The fifth is perception in Cluster #9, with bursts of 3.59. The sixth is language in Cluster #3, with bursts of 3.52. The seventh is stress in Cluster #1, with bursts of 3.42. The eight is sound disorders in Cluster #3, with bursts of 3.39. The ninth is principles in Cluster #9, with bursts of 3.15. The 10th is intervention in Cluster #9, with bursts of 3.08.

In the context of the neuropsychological assessment and diagnosis of childhood CAS, the clustering of research articles into coherent themes based on text mining and analysis techniques reveals significant patterns and trends (See Figure 3 and Table 1). The columns labelled “Label LSI”, “Label LLR”, and “Label MI” represent various analytical lenses through which the corpus of CAS literature is examined for prevalent topics and emergent themes. The “Label LSI” (Latent Semantic Indexing) column identifies overarching themes within the corpus based on the distributional semantics of the text, which can capture the context within which words are used across the dataset. The “Label LLR” (Log-Likelihood Ratio) column, on the other hand, pinpoints specific key phrases that are statistically significant within the literature, such as “developmental apraxia”, “tabby talk”, and “intellectual disability”. The “Label MI” (Mutual Information) column quantifies the strength of the association between these key phrases and the broader themes, providing insight into the most salient topics within each cluster. The connection between “Label LLR” and “Label MI” is crucial as it bridges the gap between the identification of key phrases and their relevance within the broader research themes.

The major clusters identified in the study of childhood apraxia of speech (CAS) provide a comprehensive mapping of the landscape of research and practice in this domain. Cluster #0, the largest, focuses on “Developmental Apraxia” and underscores the complex nature of speech disorders linked to developmental issues. With a focus on a broad range of studies, including the impact of fragile X syndrome, this cluster highlights the intricate link between genetic factors and speech disorders. Similarly, Cluster #1, labelled “Tabby Talk”, encompasses a significant number of studies on technological interventions, such as augmentative and alternative communication (AAC) tools, demonstrating the growing trend towards incorporating technology in therapeutic strategies. These clusters signify a shift towards integrating diverse and multifaceted approaches in diagnosing and treating CAS, reflecting both traditional and innovative methodologies.

On the other hand, Clusters like #5, “Speech Characteristics”, and #6, “Functional Characteristics”, delve into the specific attributes and impacts of CAS. These clusters explore various characteristics of speech impairments, from phonological awareness to their broader functional impacts on daily living. This highlights the necessity for targeted diagnostic tools and custom-tailored intervention plans that address both the speech-specific and functional challenges faced by individuals with CAS. The high silhouette values of these clusters suggest a strong coherence within the studies grouped under them, indicating a robust consensus on the features and implications of CAS within these areas. These insights are invaluable for developing effective assessment and diagnostic models that are both comprehensive and specific, providing a clearer path towards effective management and intervention in childhood apraxia of speech. Regarding Figure 3, it could be seen sometimes that there is an overlap in the content of the clusters and this demonstrates the content is included in more than one cluster. 

### 3.2. Thematic Analysis

Table 2 summarizes the 25 top-cited studies on the assessment of CAS, representing a foundational compendium of research which has significantly influenced the field. These seminal works are characterized by their pioneering approaches to diagnosing and understanding CAS. With studies ranging from the late 1990s to the early 2010s, the table captures the evolution of assessment techniques, from the identification of speech sound disorders and the exploration of genetic factors to the nuanced analysis of speech motor control and the development of specific intervention strategies. The collective narrative of these studies underscores the complexity of CAS as a diagnostic category and highlights the multifaceted nature of its assessment. Each entry in the table references key studies that have contributed to the establishment of diagnostic markers, therapeutic approaches, and the overall clinical understanding of CAS. The studies serve not only as a historical archive but also as a benchmark for current and future research directions, emphasizing the importance of evidence-based practice in the assessment and treatment of CAS.

Conversely, Table 3 contains the 10 most recent studies on CAS assessment, all published within the last few years, reflecting the current state of research and emerging trends in the field. This collection of studies indicates a shift towards exploring novel diagnostic biomarkers, refining intervention methods, and understanding the broader psychosocial implications of CAS. These contemporary works delve into areas such as the potential role of gluten sensitivity in CAS [37], the efficacy of motor-based treatments like dynamic temporal and tactile cueing [38], and the impact of CAS on children’s self-reported communication attitudes [39]. The table highlights the ongoing efforts to refine the diagnostic criteria and treatment efficacy, as well as a growing recognition of the importance of including a child’s perspective in the assessment process. Together, these studies represent the cutting-edge of CAS research, with a clear focus on advancing the field through innovative methodologies and a child-centric approach to intervention.
brainsci-14-00540-t002_Table 2Table 2A synthesis of the 25 top-cited studies on the diagnosis and assessment of childhood apraxia of speech.No.CitationAimFindingsDiagnostic and Assessment Measures1[40]The study aimed to distinguish children with developmental apraxia of speech (DAS) from those with speech delay (SD) by identifying reliable differences in their speech error profiles.The study supported the differentiation of suspected DAS from SD based on speech error profiles, providing evidence for the clinical utility of the DAS diagnostic category.A local ascertainment study was utilized, examining speech error profiles for diagnostic markers.2[41]The objective was to characterize patients with developmental verbal dyspraxia (DVD) and investigate its genetic underpinnings, particularly the absence of a paternal FOXP2 gene.The absence of paternal FOXP2 was identified as the cause of DVD in patients, implicating the parent-of-origin expression of FOXP2 in human speech development.Diagnostic assessment included the characterization of DVD in the context of FOXP2 gene deletions or disruptions and maternal uniparental disomy of chromosome 7.3[42]The study’s primary aim was to scrutinize the differences in speech/language and written language skills between children with suspected CAS and children with other speech sound disorders at school age.Children with suspected CAS demonstrated persistent difficulties in syllable sequencing, nonsense word repetition, and language abilities and were at risk for reading and spelling problems.Speech and language assessments, including articulation, diadochokinesis, language, reading, and spelling measures, were conducted.4[28]The study aimed to identify objective measures that could differentiate children with suspected CAS from those with other speech disorders.A combination of measures including syllable segregation and lexical stress matches achieved a high diagnostic accuracy for CAS, suggesting oral–motor examination and polysyllabic production accuracy as key diagnostic tools.The diagnostic measures included perceptual CAS features, polysyllabic picture-naming tasks, and oral–motor examinations including diadochokinesis.5[43]The study hypothesized that the speech disorder observed in individuals with galactosemia may fulfil the diagnostic criteria for CAS.The prevalence of CAS in participants with galactosemia was found to be significantly higher than in the general population, indicating a potential genetic risk factor for motor speech disorders.Participants underwent a comprehensive assessment that included cognitive, structural, sensorimotor, language, speech, prosody, and voice evaluations.6[44]The study investigated whether the core phenotype of CAS includes deficits in auditory perceptual encoding, memory, and transcoding processes.The study concluded that individuals with CAS exhibit deficits across encoding, memory, and transcoding, with the SRT showing a moderate diagnostic accuracy for identifying these deficits.The Syllable Repetition Task (SRT) was used to assess speech processing competencies including encoding, memory, and transcoding.7[16]The research aimed to delineate determiners for the differential diagnosis of developmental apraxia of speech (DAS) by analyzing the symptoms consistently used for diagnosis.Only a fraction of children referred for DAS were confirmed with the diagnosis, highlighting the necessity for distinct features in differential diagnosis.Longitudinal phonological and language evaluation data were analyzed to identify critical features for differential diagnosis.8[45]The study sought to identify the criteria speech–language pathologists use to diagnose developmental apraxia of speech (DAS).A small set of characteristics such as inconsistent productions and oral–motor difficulties were commonly used for diagnosis, demonstrating ambiguity in the criteria for DAS.A survey capturing the diagnostic criteria used by clinicians, with the identification of the characteristics most frequently associated with DAS.9[46]This research aimed to test the efficacy of a treatment targeting dysprosody in children with CAS, particularly the control of syllable durations in lexical stress.All participating children showed improvements in controlling syllable duration for lexical stress patterns, with some generalization to untreated stimuli, supporting the treatment’s efficacy.An intensive treatment trial using principles of motor learning (PML) to target syllable duration control in nonwords with varying stress patterns.10[47]The study assessed the efficacy of treatment for young children with severe CAS using a dynamic temporal and tactile cueing approach based on motor learning principles.Rapid changes in speech performance following treatment were observed, suggesting the effectiveness of intensive and frequent motor learning-based therapy.A single-subject multiple-baseline design across behaviors with continuous data collection was employed to evaluate the treatment approach.11[48]The study compared the effects of random versus blocked practice schedules in CAS treatment to evaluate their impact on speech motor learning.The results were mixed, with some children showing a preference for blocked practice and others for random practice, indicating variability in the efficacy of practice schedules.An alternating-treatment single-subject design with multiple baselines across behaviors was used to compare random and blocked practice within the dynamic temporal and tactile cueing treatment.12[49]The objective was to systematically review the treatment outcomes for children with CAS, evaluating the evidence for various therapeutic approaches.Several treatments showed positive responses, with three approaches demonstrating sufficient evidence for Phase III trials and clinical practice.A systematic review of treatment studies, including single-case experimental designs and case series, analyzed for improvement rate differences and effect sizes.13[50]The study aimed to validate the hypothesis that inappropriate stress is a reliable marker for differentiating children with DAS from those with speech delay (SD).A significant number of children with suspected DAS exhibited inappropriate stress, suggesting it as a diagnostic marker for a DAS subtype.Cross-validation of stress findings using conversational speech samples from multiple diagnostic facilities and comparison with speech delay profiles.14[51]The study aimed to link the symptoms of CAS to specific information processing deficits using computational neural modeling.The model suggested that key symptoms of CAS could be due to a reliance on feedback control caused by poor feed-forward commands.Computational modeling with the DIVA (Directions Into Velocities of Articulators) model to simulate speech production deficits in CAS.15[52]The research aimed to validate a lexical stress marker for childhood apraxia of speech (sAOS) using acoustic correlates of stress in bisyllabic word forms.The lexical stress ratio (LSR) successfully differentiated children with sAOS from those with speech delay (SD), supporting its diagnostic utility.Analysis of lexical stress task responses, with the development of a composite lexical stress ratio (LSR) as a potential diagnostic marker16[53]The aim was to investigate whether inappropriate stress patterns could serve as a diagnostic marker to differentiate suspected DAS from SD in children.Inappropriate stress was confirmed as a distinguishing feature in some children with suspected DAS, proposing it as a potential diagnostic marker.Comparative analysis of speech and prosody–voice profiles between children with suspected DAS and those with SD.17[29]The study aimed to compare the efficacy of the Rapid Syllable Transition (ReST) treatment to the Nuffield Dyspraxia Programme-Third Edition (NDP3) for children with CAS.Both ReST and NDP3 demonstrated large treatment effects, with ReST showing superior maintenance post treatment and both showing significant generalization.A randomized controlled trial evaluating articulation and prosodic accuracy before and after treatment interventions.18[54]The study explored the significance of production frequency in speech therapy for CAS and its impact on motor learning and generalization.A higher production frequency in treatment led to faster target acquisition and better generalization, emphasizing its importance in therapy for CAS.An alternating treatment AB design comparing different production frequencies in speech therapy sessions.19[55]The purpose was to deepen the understanding of the underlying deficits in DAS, particularly the planning/programming of speech movements in context, by analyzing coarticulation patterns.Children with DAS displayed more variability in coarticulation, less distinction between vowels, and idiosyncratic patterns, suggesting deficits in speech movement planning.Second formant frequency measurements were conducted in repetitions of nonsense utterances to analyze coarticulation.20[56]This study aimed to uncover the neurobiological basis of speech disorders and the potential genetic contributions, focusing on FOXP2 associations.A novel small intragenic FOXP2 deletion was identified in a child with severe motor speech disorder, suggesting a genetic basis for speech disorders, including CAS.Genetic screening for variants in the FOXP2 gene among probands with speech disorder and their families.21[57]The study aimed to examine the speech characteristics of patients treated for galactosemia and search for the presence of verbal dyspraxia.A high percentage of galactosemia patients exhibited verbal dyspraxia, indicating a specific speech disorder associated with this metabolic condition.Assessment of speech characteristics in patients treated for galactosemia to identify the prevalence of verbal dyspraxia.22[58]The research sought to determine if speech inconsistency is a unique feature of CAS and if it can differentiate CAS from speech delay.Speech inconsistency was found to be a core feature of CAS, with its measurement varying depending on the stimuli used.Assessment of speech inconsistency at phonemic and token-to-token levels using a variety of stimuli to diagnose CAS.23[20]The study aimed to use whole-exome sequencing to explore the genetic heterogeneity in CAS.The findings supported genetic heterogeneity in CAS, with several clinically reportable variants in the genes associated with CAS and related disorders.Whole-exome sequencing to identify clinically reportable variants in the genes associated with CAS in a small cohort.24[59]The study’s goal was to define the phenotype associated with 12p13.33 microdeletion, which includes ELKS/ERC1, and investigate its link with CAS.Deletion in the 12p13.33 locus, including the ELKS/ERC1 gene, was associated with CAS, suggesting a genetic basis for speech sound disorders.Array CGH to identify subtelomeric and interstitial rearrangements, with the clinical assessment of speech and neurobehavioral development.25[60]The purpose was to examine the impact of feedback frequency on treatment for CAS and its role in motor learning and treatment efficacy.Mixed results were observed, indicating that the benefits of feedback frequency alterations in CAS treatment may vary among individuals.An alternating-treatment single-subject design with multiple baselines across behaviors to compare high-frequency and low-frequency feedback.
brainsci-14-00540-t003_Table 3Table 3A synthesis of the 10 most recent 10 cited studies on the diagnosis and assessment of childhood apraxia of speech.No.CitationAimFindingsDiagnostic and Assessment Measures1[37]This study aimed to investigate the relationship between gluten sensitivity and neurotransmitter alterations (glutamate and GABA) in children with CAS.Children with CAS showed higher levels of antigliadin IgA and glutamate, with a lower level of GABA, indicating a possible diagnostic marker for CAS relating to gluten sensitivity and neurotransmitter imbalance.Plasma levels of antigliadin IgA, glutamate, and GABA were determined by enzyme-linked immunosorbent assay (ELISA) in children with CAS, DLD, and NT.2[61]The purpose was to identify early indicators of CAS in late talkers and provide general guidelines for intervention.Varying responses to intervention among late talkers suggested potential early indicators of CAS, influencing diagnostic decisions and the early implementation of motor learning principles in treatment.Implementation of the Target Word program and focused stimulation, along with parental guidance for stimulating language development.3[62]The study aimed to validate the Multilevel Word Accuracy Composite Scale (MACS) as a novel measure of speech production in children with CAS.The MACS showed positive correlations with established speech accuracy measures and demonstrated moderate-to-excellent reliability among expert raters and SLPs.Utilization of the MACS to rate speech production accuracy, with reliability and validity tested against established measures.4[63]This study explored SLPs’ and OTPs’ perspectives on comorbidity of CAS and sensory processing disorder (SPD) and the effectiveness of interprofessional approaches.The findings emphasized the need for a unified clinical language and interprofessional approaches to improve treatment for children with CAS and comorbid SPD.Reflexive thematic analysis of responses from SLPs and OTPs regarding the comorbid presentation and treatment of CAS and SPD.5[39]The study investigated the self-reported communication attitudes of children with CAS and their implications for assessment and intervention.Older children with CAS exhibited more negative communication attitudes, highlighting the importance of including child self-report measures in research and therapy.Administration of validated communication attitude questionnaires to children with CAS enrolled in an intensive speech-focused intervention.6[38]The paper outlined a protocol for a randomized controlled trial comparing the efficacy of DTTC treatment with varying dose frequencies in children with CAS.The study aimed to provide evidence for the optimal treatment schedule for DTTC by examining treatment outcomes at different dose frequencies.Randomized controlled trial protocol comparing low versus high dose frequency of DTTC treatment in children with CAS.7[64]This research examined manual rhythmic sequencing skills in children with hx/CAS compared to typically developing children.Children with hx/CAS showed deficits in manual rhythmic sequencing, providing support for a domain-general cognitive mechanisms account of rhythmic deficits in CAS.Assessment of manual rhythmic sequencing skills using clapping and tapping tasks in children with hx/CAS and TD.8[65]The study described the development of self-made gestures as an adaptive communication strategy in an individual with CAS.The individual with CAS developed self-made gestures for communication that eventually subsided as verbal communication improved, demonstrating an adaptive strategy for coping with expressive challenges.Observational case study of the development and use of self-made gestures in an individual with CAS.9[66]The study aimed to compare the outcomes of two motor-based treatments, ReST and ultrasound biofeedback, for school-age children with CAS.Both treatments were equally effective, showing significant improvements in speech sound accuracy and prosody, with treatment effects maintained at the 1-month follow-up.A pilot randomized control trial comparing ReST and ultrasound biofeedback treatment outcomes in children with CAS.10[67]The research aimed to examine the clinical practice of CAS in Hong Kong to inform future research and evidence-based practice.The study found that local SLPs’ understanding of CAS was limited and that evidence for the assessment, diagnosis, and treatment of Cantonese speakers with CAS was lacking.Web-based survey study on the knowledge and experience of Hong Kong pediatric SLPs with CAS, including assessment, diagnosis, and treatment practices.


### 3.3. Categorization of Existing Diagnosis and Assessment Measures of Childhood Apraxia of Speech

Table 4 organizes the diagnostic and assessment measures for CAS into categories, with each category reflecting a distinct aspect of relevance to the diagnosis and assessment of CAS. Sample studies are selected to represent each category and illustrate the various approaches taken to understand, diagnose, and assess CAS in the research literature.

The multifaceted nature of CAS necessitates a diagnostic and assessment approach that is both comprehensive and nuanced. The neurobiological underpinnings of CAS, for instance, have been a focal point of recent research endeavors [37]. Such studies have identified genetic and biochemical markers that may be indicative of CAS, with correlations drawn between heightened levels of specific antibodies, neurotransmitters, and genetic mutations, and the manifestation of the disorder [41]. This burgeoning domain offers a promising avenue for early and accurate diagnosis, as it provides clinicians with objective measures that may differentiate CAS from other speech and language disorders. The quest for these biological markers is driven by the goal of establishing a more precise diagnostic framework that can underpin targeted interventions.

In addition to the biological perspective, the assessment of speech motor control has been recognized as a cornerstone in the diagnosis of CAS. Researchers have meticulously delineated the speech production deficits characteristic of CAS, with a particular focus on the precision and consistency of speech motor plans [40]. Studies employing tasks that challenge the speech motor system have revealed significant discrepancies between children with CAS and their typically developing peers, particularly in tasks requiring the coordination of complex articulatory movements [64]. Moreover, perceptual speech features have been scrutinized to distinguish CAS from other speech sound disorders, utilizing expert judgment and perceptual rating scales to capture the subtle nuances of speech that are often disrupted in CAS [29,62]. These measures, although reliant on the subjective discernment of clinicians, add depth to the diagnostic process, affording a richer understanding of the disorder’s impact on speech production.

The complexity of CAS diagnosis is further compounded when considering the disorder’s comorbidity with other developmental conditions, such as sensory processing disorder (SPD). The interplay between CAS and SPD has prompted research into the benefits of an interprofessional approach to both assessment and treatment [63]. Such collaboration underscores the need for a unified clinical language and fosters a holistic approach to addressing the multifarious challenges faced by children with CAS. Furthermore, the cultural and linguistic context within which CAS is assessed cannot be overstated, as it shapes both the interpretation of symptoms and the subsequent intervention strategies [59,67]. This is of particular salience in multilingual societies, where the nuances of different languages and cultural norms must be carefully negotiated to ensure equitable and effective diagnostic and therapeutic practices.

In short, the literature presents a constellation of categories essential for the comprehensive assessment and diagnosis of CAS. These categories span the biological, motoric, perceptual, and socio-cultural dimensions of the disorder, each contributing a vital piece to the intricate puzzle of CAS. The synthesis of these domains offers a promising scaffold upon which future research and clinical practices can be built, with the aim of enhancing the lives of those affected by this challenging speech disorder.

### 3.4. A Conceptual Model Describing and Prescribing Diagnosis and Assessment of Childhood Apraxia of Speech

The conceptual model presented in Figure 4 aims to encapsulate and advance the existing measures for diagnosing and assessing CAS by integrating insights from the most cited and recent studies within the field. Central to this model is a multi-tiered approach, aligned with the categories identified, that converges on a comprehensive understanding of CAS through the prisms of neurobiology, speech motor control, perceptual features, auditory processing, prosody and stress, and personal and community perspectives.

At the foundation of the model lies the neurobiological markers that inform the genetic and biochemical landscape of CAS. This tier involves the assessment of genetic predispositions, such as FOXP2 mutations, and the measurement of neurotransmitter levels that influence speech and language processing [37,41]. This tier serves as a grounding point, providing objective data which may guide the early identification and diagnosis of CAS. Building upon the neurobiological framework, the second tier focuses on the mechanisms of speech motor control. This includes the assessment of articulatory kinetics, coordination, and sequencing skills necessary for fluent speech production. Motor control measures such as the accuracy of syllable production and oral–motor examinations are critical [40,64]. This tier also integrates motor learning principles, observing how children with CAS acquire and generalize speech motor patterns in response to various treatment modalities [38,48].

The third tier examines perceptual speech features and auditory processing capabilities. Here, perceptual assessments such as the Multilevel Word Accuracy Composite Scale (MACS) and auditory processing tasks help delineate the subjective aspects of speech that are often disrupted in CAS [42,62]. Perceptual features, including prosody and stress patterns, are essential in differentiating CAS from other speech sound disorders [43]. The fourth tier of the model considers personal factors, including self-reported communication attitudes and parent reports, to gauge the functional impact of CAS on daily life [39]. Additionally, an analysis of comorbidities, such as sensory processing challenges, helps refine the diagnosis and tailor interventions [63].

The apex of the model is characterized by the cultural and linguistic dimensions of CAS assessment. This tier recognizes the diversity in speech patterns across languages and cultures and emphasizes the need for diagnostic measures that are culturally and linguistically sensitive (e.g., ASRS, DEMSS) [59,67]. The model ends with a dynamic integration of these tiers, with feedback loops which allow for the continual refinement of the diagnostic process. As new research emerges, particularly in genetic and neurobiological markers, these findings are fed back into the model to recalibrate and enhance the assessment measures. Similarly, data from personal factors and cultural dimensions can inform the interpretation and application of neurobiological and motor control assessments.

Our model is intrinsically linked to the framework of The International Classification of Functioning, Disability, and Health (ICFDH) by incorporating its comprehensive view of health that includes both functioning and contextual factors [68] (See Figure 5). The ICF framework emphasizes a multidimensional approach to health, encapsulating body functions and structures, activity and participation, and contextual factors. Our model aligns with this by addressing the neurobiological underpinnings of CAS, which corresponds to the “Body Functions and Structures” component of the ICF. It further evaluates speech motor control and perceptual features, paralleling the “Activity and Participation” aspect, as it examines how these factors impact daily communication and interaction. Additionally, our model considers personal and community perspectives, akin to the “Contextual Factors” in the ICF, recognizing the influence of environmental and personal contexts such as cultural and linguistic diversity on CAS assessment and intervention. By integrating these elements, our model not only advances the diagnosis and understanding of CAS but also aligns with the ICF’s holistic view of health and functioning, promoting a more personalized and contextually relevant approach to care.

## 4. Discussion

This study aimed to integrate findings from cluster analysis and thematic analysis to articulate a comprehensive understanding of the diagnosis and assessment of CAS. The main findings revealed from the cluster analysis indicated the prevalent research themes and identified gaps within the current literature. The thematic analysis of the top-cited and most recent studies provided a deeper insight into the prevailing methodologies and diagnostic criteria. A model was developed, synthesizing the quantitative and qualitative findings to describe and prescribe refined measures for diagnosing and assessing CAS. This model underscores the necessity for evidence-based practice, drawing from the reliability and validity of tools such as the Apraxia of Speech Rating Scale (ASRS) [31] and the Dynamic Evaluation of Motor Speech Skill (DEMSS) [34], and emphasizes the importance of further validation through maintenance and generalization measures [49].

Key findings from the synthesized studies in Table 2 and Table 3 include the differentiation of CAS from other speech disorders such as SD based on speech error profiles, the identification of genetic underpinnings such as FOXP2 gene deletions or disruptions, and the association of CAS with galactosemia and gluten sensitivity. Several studies emphasize the importance of specific features for a differential diagnosis, including inappropriate stress patterns, speech inconsistency, and deficits in encoding, memory, and transcoding processes. The efficacy of various treatments is also explored, with motor learning-based approaches such as ReST and DTTC showing promising results. The tables also highlight the need for a unified clinical language, interprofessional approaches, and the inclusion of child self-report measures in research and therapy. The study of manual rhythmic sequencing skills and the development of self-made gestures as an adaptive communication strategy provide additional insights into the cognitive and expressive aspects of CAS.

The discourse surrounding CAS has been enriched by the utilization of innovative tools that facilitate the accurate identification and description of speech characteristics. The ASRS has emerged as a reliable and valid instrument, capturing the nuances of apraxic speech [31]. Similarly, the DEMSS adds value to a differential diagnosis, effectively distinguishing CAS from other severe speech impediments [34]. These tools align with previous findings, which have stressed the need for objective measures which reliably differentiate CAS from other speech disorders [28]. Concurrently, the advent of automated tools like “Tabby Talks” [32] and deep neural network-based classification tools [69] reflect an evolving landscape wherein technology aids in the complex process of CAS assessment.

The application of neurocomputational modeling, as discussed by [30], represents a forward-thinking approach to unravelling the complexities of CAS at a neuromotor level. This analytical tool allows researchers to simulate and predict the impacts of specific neuromotor deficits on speech production, offering a potential blueprint for the development of targeted therapies. This innovative method aligns with calls for a more nuanced understanding of CAS and its management, as echoed by [70], who highlight the necessity for an array of diagnostic features and measurement procedures in research studies. It is through such comprehensive and multifaceted approaches that the field can aspire to tailor interventions to the unique profiles of individuals with CAS.

In synthesizing the insights from the cluster and thematic analyses, this study presents a model that encapsulates the current understanding of CAS while also paving the way for future research and clinical practice. The integration of technological tools like “Tabby Talks” and deep neural network-based classifiers [32,69] with established assessment measures provides an exciting trajectory for enhancing the precision and efficiency of CAS diagnosis and treatment. These tools, however, need to be considered within the broader context of individual variability and the multifactorial nature of CAS. As the field progresses, it will be crucial to balance technological innovation with personalized care, ensuring that each child with CAS receives the most effective, evidence-based, and individualized intervention possible. The model proposed in this study, therefore, not only crystallizes current knowledge but also serves as a catalyst for ongoing inquiry, aiming for a future where the diagnosis and treatment of CAS are optimized for the best possible outcomes.

### 4.1. Limitations

This study, while comprehensive in its approach to integrating quantitative and qualitative analyses, is not without its limitations. One of the primary constraints lies in its reliance on published literature, which may be subject to publication bias, as studies with significant findings are more likely to be published than those with null results. Additionally, the exclusion of non-English language studies may omit valuable insights from research conducted in other linguistic and cultural contexts, potentially limiting the generalizability of the findings. Furthermore, the inherent complexity of CAS, with its varied presentations, may not be fully captured by the studies included in the analysis, which could lead to an oversimplified representation of the disorder. The technological tools identified, such as “Tabby Talks” and deep neural network-based classifiers, while promising, may have not yet undergone sufficient clinical validation to confirm their effectiveness across diverse populations and settings. These limitations highlight the need for ongoing research and the continuous refinement of the model proposed in this study.

### 4.2. Implications

The implications of this study extend into clinical, educational, and research domains, emphasizing the need for dynamic and evidence-based practices in the diagnosis and treatment of CAS. Clinically, this study reinforces the importance of using a combination of validated tools and expert judgment to ensure an accurate diagnosis and tailor interventions to individual needs. Educators and therapists may find the model proposed here useful in structuring their assessment strategies and intervention plans. For researchers, this study underscores the necessity for further empirical investigations that include maintenance and generalization measures, especially in underrepresented languages and cultures. It also encourages the pursuit of interdisciplinary collaboration to enhance the understanding of CAS and its management. The technological advancements identified call for rigorous testing in real-world clinical settings to establish their efficacy and potentially revolutionize the way in which speech–language pathologists assess and treat CAS.

## 5. Conclusions

This study synthesizes a wealth of research to form a model that offers a comprehensive view of the diagnosis and assessment of CAS. The model serves as both a reflection of the current state of understanding and a guide for future endeavors in this domain. It underscores the intricate interplay between evidence-based practice, technological innovation, and individualized care in the management of CAS. As the field continues to evolve, this model provides a foundation upon which to build more nuanced and effective approaches to intervention, ensuring that individuals with CAS receive the highest quality of care. The goal of this research is to empower speech–language pathologists with the tools and knowledge necessary to facilitate optimal communication outcomes for children with CAS across diverse contexts.

## Figures and Tables

**Figure 1 brainsci-14-00540-f001:**
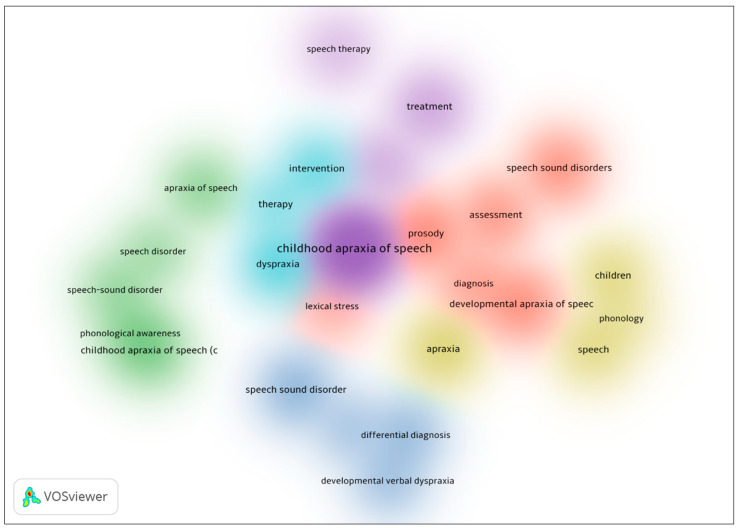
Density of visualization of word co-occurrence in childhood apraxia of speech.

**Figure 2 brainsci-14-00540-f002:**
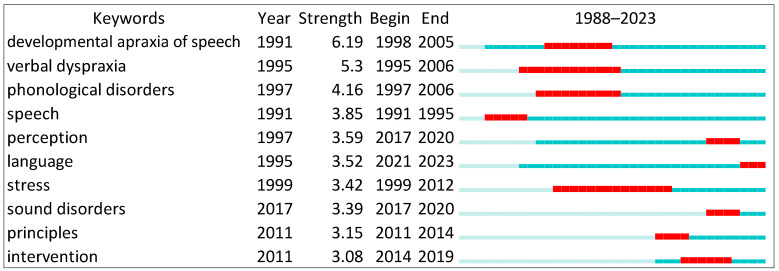
Top 10 keywords with the strongest citation bursts.

**Figure 3 brainsci-14-00540-f003:**
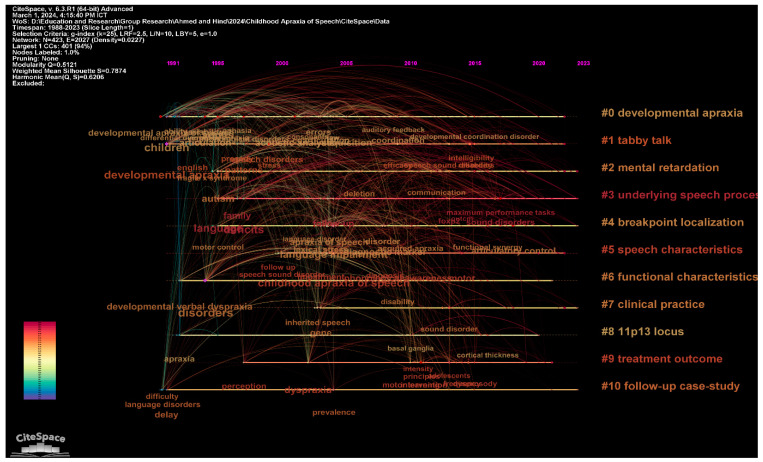
Visualization of the top clusters in childhood apraxia of speech.

**Figure 4 brainsci-14-00540-f004:**
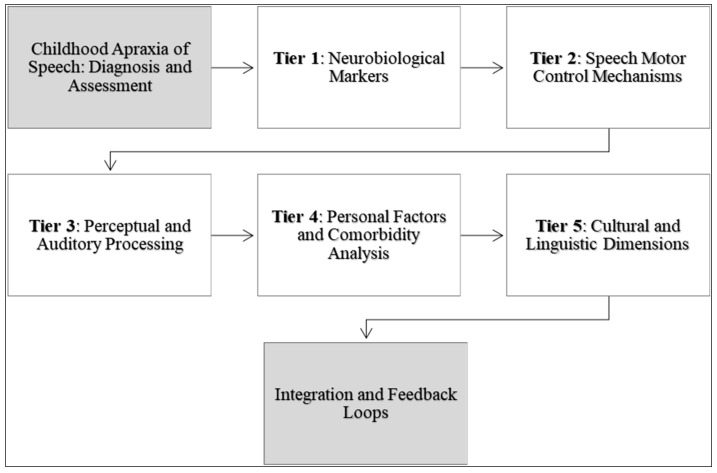
A descriptive and prescriptive framework for advancing the diagnosis and assessment of childhood apraxia of speech.

**Figure 5 brainsci-14-00540-f005:**
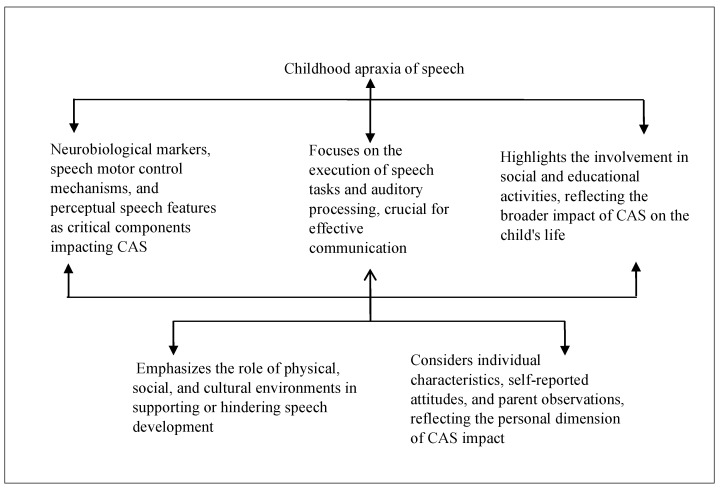
Diagnosis and assessment of childhood apraxia of speech according to the ICFDH framework.

**Table 1 brainsci-14-00540-t001:** Summary of the 10 largest clusters.

Cluster ID	Size	Silhouette	Label (LSI)	Label (LLR)	Label (MI)	Average Year
0	63	0.725	childhood apraxia	developmental apraxia	speech part ii (1.91)	2002
1	48	0.696	childhood apraxia	tabby talk	AAC intervention (0.88)	2009
2	45	0.779	childhood apraxia	mental retardation	childhood apraxia (0.22)	2010
3	45	0.739	childhood apraxia	underlying speech processes	pathogenic role (0.72)	2015
4	41	0.832	childhood apraxia	breakpoint localization	pathogenic role (0.33)	2009
5	35	0.79	childhood apraxia	speech characteristics	pathogenic role (1.11)	2012
6	31	0.755	childhood apraxia	functional characteristics	pathogenic role (0.25)	2009
7	26	0.92	childhood apraxia	clinical practice	childhood apraxia (0.25)	2012
8	26	0.933	developmental verbal dyspraxia	11p13 locus	childhood apraxia (0.27)	2007
9	24	0.819	childhood apraxia	treatment outcome	tablet-based delivery (0.3)	2013

**Table 4 brainsci-14-00540-t004:** A categorization of the 25 top-cited and 10 most recent studies on the diagnosis and assessment of childhood apraxia of speech.

No.	Category	Elaboration	How Studies Accounted for the Category	Sample Studies
1	Neurobiological Markers	Examining the biochemical and genetic underpinnings of CAS to aid in diagnosis.	Studies measured specific biomarkers, such as neurotransmitter levels or gene mutations, which may be linked to the etiology of CAS.	[37,41]
2	Speech Motor Control	Assessing speech production and motor planning as a diagnostic measure for CAS.	Utilized tasks and tools that measure the accuracy and consistency of speech motor control, often comparing performance between CAS and typically developing children.	[40,64]
3	Perceptual Speech Features	Identifying perceptual characteristics of speech that are indicative of CAS.	Studies relied on expert judgment and perceptual rating scales to distinguish CAS from other speech disorders.	[28,29,62]
4	Auditory Processing	Exploring the role of auditory perception and encoding in CAS.	Investigated how children with CAS perceive and encode auditory information, and whether deficits in these areas can be diagnostic markers.	[42,44]
5	Prosody and Stress Patterns	Focusing on the rhythmic and intonational aspects of speech that are often disrupted in CAS.	Studies examined specific prosodic features, such as lexical stress, to differentiate CAS from other speech disorders.	[43,64]
6	Parent- and Self-Report Measures	Using subjective reports from parents and individuals with CAS to inform diagnosis and treatment.	Studies included questionnaires or interviews that captured the lived experience and communicative challenges faced by those with CAS.	[39,52]
7	Intervention Response	Assessing changes in speech following specific therapeutic interventions as a diagnostic tool.	Evaluated how children with suspected CAS respond to targeted speech therapy as a means of confirming diagnosis.	[61,66]
8	Motor Learning and Generalization	Observing the ability to learn and generalize motor patterns as characteristic of CAS.	Studies focused on the application of motor learning principles in assessment and treatment to determine the ability to generalize learned speech patterns.	[38,60]
9	Comorbidity Analysis	Investigating the co-occurrence of CAS with other developmental disorders.	Explored how the presence of other conditions, such as sensory processing disorder, may influence the diagnosis and treatment of CAS.	[53,63]
10	Cultural and Linguistic Considerations	Considering the influence of language and culture on the assessment and diagnosis of CAS.	Studies acknowledged the need for culturally and linguistically appropriate diagnostic measures for CAS, especially in multilingual contexts.	[59,67]

## Data Availability

The raw data supporting the conclusions of this article will be made available by the authors upon request.

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
