# Peer review of "Childhood Apraxia of Speech: A Descriptive and Prescriptive Model of Assessment and Diagnosis"

_brainsci, 2024, doi:10.3390/brainsci14060540_

Round 1
Reviewer 1 Report
Comments and Suggestions for Authors
The present study used both quantitative and qualitative analysis on previous literature on CAS in terms of its assessment and diagnosis, leaving some potential contribution to this field. However, the authors should address the following concerns.
1. The authors should discuss the similarity and difference between diagnostic and assessment. To some readers, the two could be interchangeable.
2. The research gap should be introduced and delineated in depth before the purpose of the present study. I am not persuaded by this version.
3. Why did the authors only included WOS and Scopus as their database. I suggest, PubMed, PsyInfo, and EBSCO should also be included in the databases for literature searching.
4. The authors mentioned 25 most cited and 10 most recent articles. How is number determined? This seems very arbitrary.
5. For the results presentation, the authors mostly repeated the table and figure’s information, so I suggest the authors provide some summary and further explanation of the results with a plain writing style, for some readers might not be familiar with Silhouette and other indexes.
6. The authors delineated a model, but I am not very persuaded. How are tiers determined and why there are tiers? What are the relationship between the tiers, and how the order of the tiers is decided?
7. The discussion is relatively short, and I suggest the authors should re-organize the discussion into different sections with headings, to help readers understand. The current version is not clear enough.
Comments on the Quality of English Language
The present version seems fine, but there is still room to improve.
Author Response
"Please see the attachment."

Reviewer 2 Report
Comments and Suggestions for Authors
The bibliography and in-text references must be organized according to journal standards and the bibliography must be edited accordingly.
The abstract is long in word count and keywords are more extended phrases than index words.
The text in the introduction should refer to the DIVA and GODIVA models and the general principles of diagnosis and comorbidity related to childhood apraxia so that the theoretical framework is complete and then harmonized with the resulting clusters and the frequency of occurrence of accompanying features, the factors surrounding the disorder, the symptoms and so on.
It is good to strengthen the material of the tests - tests for diagnosis and assessment of apraxia, to add literature from existing systematic reviews on the subject, possibly also from meta-analyses to strengthen the theoretical framework in the fields of tools and diagnosis and to be as well-documented and connected as possible your text later in the conversation.
e.g
Murray, E., Iuzzini-Seigel, J., Maas, E., Terband, H., & Ballard, K. J. (2021). Differential diagnosis of childhood apraxia of speech compared to other speech sound disorders: A systematic review. American Journal of Speech-Language Pathology, 30(1), 279-300.
Oliveira, A. M. D., Nunes, I., Cruz, G. S. D., & Gurgel, L. G. (2021). Methods of assessing childhood apraxia of speech: systematic review. Audiology-Communication Research, 26, e2524.
Chilosi, A. M., Podda, I., Ricca, I., Comparini, A., Franchi, B., Fiori, S., ... & Santorelli, F. M. (2022). Differences and commonalities in children with childhood apraxia of speech and comorbid neurodevelopmental disorders: A multidimensional perspective. Journal of personalized medicine, 12(2), 313.
The methodology is missing something very important. It should be noted in detail how the review was done, your criteria, the keywords combined according to PICO possibly, and the inclusion and exclusion criteria of the articles in even more detail.
Within the tables, there is information on treatment protocols and models of treatment and evaluation e.g. CGS, DTTC, SRT, PML, MACs, and others which should even briefly be mentioned in your theoretical background.
Edit the formatting and font size so that it is the same throughout the text following the journal’s standards.
Author Response
"Please see the attachment."

Round 2
Reviewer 2 Report
Comments and Suggestions for Authors
I confirm that all my suggestions have been taken into account, and all the required changes and modifications have been made in the text strengthening all the fields of the study. Everything requested has been addressed.
Author Response
Thank you so much and we are glad that our revisions were consistent with your expectations.